# A Two-Hit Model of Executive Dysfunction: Simulated Galactic Cosmic Radiation Primes Latent Deficits Revealed by Sleep Fragmentation

**DOI:** 10.3390/life15111717

**Published:** 2025-11-06

**Authors:** Richard A. Britten, Ella N. Tamgue, Paola Arriaga Alvarado, Arriyam S. Fesshaye, Larry D. Sanford

**Affiliations:** 1Department of Radiation Oncology, Eastern Virginia Medical School, Macon and Joan Brock Virginia Health Sciences at Old Dominion University, Norfolk, VA 23507, USA; tamgueen@odu.edu (E.N.T.); alvarapa@odu.edu (P.A.A.); fesshayas@odu.edu (A.S.F.); 2Department of Biomedical and Translational Sciences, Macon and Joan Brock Virginia Health Sciences at Old Dominion University, Norfolk, VA 23507, USA; sanforld@odu.edu; 3Center for Integrative Neuroscience and Inflammatory Diseases, Macon and Joan Brock Virginia Health Sciences at Old Dominion University, Norfolk, VA 23507, USA

**Keywords:** space radiation, sleep fragmentation, decision making, decision bias, impaired updating

## Abstract

Future Artemis-class missions to Mars will expose astronauts to prolonged space radiation (SR), sleep disruption, and operational demands requiring greater autonomy, placing decision making and executive function at heightened risk. Both SR and sleep fragmentation (SF) independently impair cognition, yet their combined effects remain poorly understood. Using the Associative Recognition Memory and Interference (ARMIT) task, we assessed cognitive performance in male rats exposed to 10 cGy of Galactic Cosmic Ray simulation (GCRsim), SF, or both. Under well-rested conditions, GCRsim-exposed rats exhibited overt deficits in the C.1.2 stage, performing at chance when reinforcement contingencies shifted, consistent with impaired cognitive flexibility. In contrast, high-performing GCRsim-exposed rats that initially performed comparably to Sham s revealed latent deficits following a single night of SF. Specifically, the SF-induced loss of C.1.3 performance was accompanied by perseverative errors (persistently selecting outdated cues despite negative feedback), reflecting impaired attentional control and decision updating. Sham s maintained stable performance after SF. These findings support a two-hit vulnerability model in which SR primes corticostriatal and frontoparietal networks for collapse under subsequent sleep disruption. Operationally, this suggests that astronauts may display either persistent or stress-induced deficits, with both modes threatening mission success. Identifying mechanisms of such vulnerabilities is essential for countermeasure development.

## 1. Introduction

The upcoming long-duration Artemis missions to Mars will present several new challenges for astronauts that may require the adoption of some new operational approaches. For a variety of reasons, astronauts will have to act more autonomously than on previous missions; thus, decision-making competency will be of increased operational significance. Decision-making is a complex process involving the rapid assessment of problems, weighing options, and selecting the most appropriate action after considering potential risks and benefits, processes that are highly dependent upon the efficient utilization of several executive functions. Executive functions, decision making and situational awareness are critical determinants of successful extravehicular activities (EVAs) task completion, including both routine operations and unexpected emergencies (incapacitated crew member rescue) [1]. These studies also recognized that stress and inadequate sleep increased the issues that astronauts experienced when using these cognitive processes.

For various reasons, astronauts frequently get insufficient restful sleep [2]. Sleep loss can reduce cognitive performance in many ways, including slower response times [3,4,5,6,7,8], increased lapses in attention [8,9,10,11,12,13,14] and memory decrements [6,7,14,15,16,17,18,19]. Sleep perturbations also increase error rates [8,20] while decreasing the ability to detect and correct errors [21,22]. Furthermore, inadequate sleep decreases inhibitory control [23] and increases risk-taking behaviors [24]. It is thus not surprising that decision-making, particularly in situations with uncertain outcomes and imperfect information (emergency response, disaster management, military encounters), is severely compromised following sleep perturbation [21,25,26,27]. The unique RIDGE (Radiation, Isolation, Distance from Earth, Gravity, and Hostile/Closed Environments) context on long-duration space missions is likely to increase the frequency and severity of sleep perturbations.

Astronauts on deep space missions will be continuously exposed to space radiation (SR) with an annual dose exposure of ~13 cGy [28,29]. In ground-based studies, exposure to <15 cGy of SR impairs executive function and cognitive flexibility performance across both sexes [30,31,32,33,34,35,36,37,38,39,40,41,42]. However, not all individuals develop SR-induced cognitive impairment, with most studies reporting that ~30% of rats will experience significant loss of performance, especially in attention-dependent tasks [30,31,39,43,44]. If similar impairments were to occur in humans, the operational consequences could be significant. For example, military units are considered non-functional if 30% of their members are “unfit for duty” [45].

At certain times during deep space missions, Artemis astronauts will have to contend with the combined impacts of SR and sleep loss. While each of these space flight stressors independently impacts cognitive performance, there appear to be unique performance decrements induced after exposure to combined stressors [37,40,41]. In those studies, approximately 30% of rats have impaired attentional set shifting (ATSET) performance following SR exposure (these rats can be classified as “SR-Sensitive”). Approximately half of “SR Refractory” rats (those that had perfect performance when normally rested) developed “latent” performance decrements after a single night of fragmented sleep (SF). These rats can be classified as “conditionally SR sensitive” rats.

These results might seem predictable, as inadequate restorative sleep is well known to impair cognitive functioning (e.g., [46,47,48]). However, the notable finding here is that SF disrupts ATSET performance primarily in animals previously exposed to SR, implying that SR sensitizes the brain to the adverse effects of sleep disruption. Furthermore, the observed performance impairments under combined SF and SR conditions are largely restricted to ATSET phases that demand sustained attention or cognitive flexibility (i.e., reversal and set-shifting stages). In contrast, SR exposure alone seldom produces comparable deficits when animals are tested in a well-rested state [30,31,37,39,41,42,49]. This pattern of selective, “latent” SF-related impairment [37,40,41] suggests that the higher cognitive demands of these stages render them particularly vulnerable to the combined impact of SR and SF.

The Associative Recognition Memory and Interference Touchscreen (ARMIT) task was designed to assess the ability of rats to rapidly solve complex problems under high cognitive load. Specifically, it challenges their capacity to multitask, such as solving a problem while managing interference from previous trials (i.e., remembering across multiple trials) [36]. The ARMIT task builds upon elements of the Mnemonic Similarity Task, a behavioral paradigm originally designed to test pattern separation in humans [50].

ARMIT provides insight into several cognitive domains, including working memory integrity, susceptibility to interference, attentional filtering (the ability to ignore irrelevant information, a function closely linked to prefrontal cortex integrity), and executive functions such as interference resolution and cognitive control. Both attentional filtering and working memory integrity are essential components of Level 1 Situational Awareness (perception) and Level 2 Situational Awareness (comprehension).

We have previously reported that exposure of female rats to 10 cGy GCRsim significantly reduced performance in the ARMIT task [30]. The “Simplified 5-ion” GCRsim beam approximates the primary and secondary SR fields that the organs of Astronauts will be exposed to during the voyage to Mars [51]. The present study has investigated the incidence, nature and severity of ARMIT impairment in male rats exposed to 10 cGy GCRsim and SF (either in isolation or combined).

## 2. Materials and Methods

### 2.1. Regulatory Compliance

All animal procedures were reviewed and approved by the Institutional Animal Care and Use Committees of Eastern Virginia Medical School (EVMS, Norfolk, VA, USA) and Brookhaven National Laboratory (BNL, Upton, NY, USA). All protocols conformed to the standards outlined in the Guide for the Care and Use of Laboratory Animals (8th Edition, National Research Council).

### 2.2. Rat Demographics, Husbandry, Experimental Timeline, and Exercise Regimen

Male Wistar rats (Hla^®^(WI)CVF^®^; Hilltop Lab Animals, Inc., Scottsdale, PA, USA) were used for all experiments. Upon arrival at EVMS, animals were approximately 3 months old and weighed about 250 g. A detailed overview of the experimental schedule relative to the time of space radiation (SR) exposure is provided in Table 1. At the time of irradiation, the rats were between 5 and 6 months of age, corresponding to an estimated human age equivalent of ~18 years [52].

Following arrival, rats were pair-housed in individually ventilated cages (Green Line Techniplast, Buguggiate, Italy) under a reversed 12:12 h light–dark cycle with unrestricted access to Teklad 2014 chow. After a one-week acclimation period, animals were implanted subcutaneously with RFID transponders (ID-100US, Electronic Devices, Santa Barbara, CA, USA) and began a mild treadmill exercise program (30 min at 25 m/min, twice per week). Exercise sessions were performed during the dark phase using a Panlab LE8710RTS five-lane treadmill (Harvard Apparatus, Holliston, MA, USA). This regimen corresponds to a light aerobic training protocol [53].

A total of 47 rats were transported to BNL for exposure to simulated space radiation.

### 2.3. Irradiation Procedure

At BNL, rats were housed under the same environmental and husbandry conditions as at EVMS. After at least one week of acclimation, animals were placed in ventilated, custom-built “rat hotel” holders made of red plastic. Twenty-one animals received a 10 cGy whole-body dose of a “Simplified 5-ion” GCRsim at the NASA Space Radiation Laboratory (NSRL) (https://www.bnl.gov/nsrl/userguide/SimGCRSim.php, accessed on 28 August 2025). This beam provides a uniform dose distribution and consists of the following components (delivered in the order shown): protons (1 GeV/n): 35%; ^28^Si (600 MeV/n: 1%; ^4^He (250 KeV/n): 18%; ^16^O (350 MeV/n): 6%; ^56^Fe (600 KeV/n): 1%; and protons (250 MeV/n): 39%). The radiation is delivered in the same sequence in which the ions are listed.

Dose calibration followed established NSRL procedures [54]. The rats were approximately 7 months old at exposure.

A separate group of 26 Sham controls was placed in identical holders but remained in the preparation room during irradiation. One week later, all animals were returned to EVMS and maintained under the same conditions described above.

### 2.4. Associative Recognition Memory and Interference Touchscreen Task (ARMIT) Screening

To enable comparison with other SR-related executive function studies, ARMIT testing was conducted approximately 12 ± 2 weeks post-irradiation (~10 months of age). Three days before testing, rats were placed on a restricted feeding schedule (~6 g of chow per day) to maintain body weight at ~85% of baseline. Behavioral testing was performed during the dark phase using Bussey–Saksida touchscreen chambers (Model 80604; Lafayette Instruments, Lafayette, IN, USA). Testing for each animal began approximately two hours into the dark cycle (Zeitgeber T + 2) and occurred at a consistent time across sessions.

#### 2.4.1. Stimulus Response Training (SRT)

SRT consists of six stages (Habituation (HAB), SRT15, SRT4, SRT1, SRT1-Fast (SRT1-F) and SRT1-Fast with punishment (SRT1-FP)).

During HAB, rats learned to retrieve sugar pellet rewards (20 mg, unflavored; Bio-Serv, Flemington, NJ, USA) from the food tray. Animals consuming all five pellets within a 30 min session advanced to the next stage. Up to three days were allowed for criterion completion; rats failing to meet the criterion were excluded from further training.

Subsequent stages were conducted daily with a maximum of 50 trials per session. Advancement required meeting the stage-specific performance criteria listed in Table 2. Rats not achieving the criterion within the designated number of sessions were removed from the study.

In SRT15, any touch to one of 15 illuminated holes (3 × 5 grid) delivered a reward. A transparent shield with 35 mm openings prevented accidental screen touches.

In SRT4, reward delivery required touching one of four illuminated holes arranged in a 2 × 2 block, randomly repositioned after each response.

In SRT1-Timed, only one illuminated hole was presented; the rat had 30 s to respond before a timeout.

In SRT1-Fast, the response window was shortened to 10 s.

Finally, in SRT1-FP, incorrect selections of unlit holes triggered an aversive bright light and a 30 s timeout.

#### 2.4.2. ARMIT Performance Evaluation

Each ARMIT trial was separated by a 5 s inter-trial interval (ITI). Initially, rats were tested on the C.1.1 configuration, where holes 2 and 4 were illuminated simultaneously; choosing either produced a reward (Figure 1). Animals completed this version for five consecutive days (50 trials/day, 30 s response window) to establish a stable memory of the rewarded pattern.

Premature touches during the ITI were recorded and resulted in restarting at C.1.1. Selecting an unlit hole had no programmed effect, whereas omission (no response within 30 s) triggered a 5 s light timeout before restarting the sequence.

On the following day, animals were exposed sequentially to the C.1.1, C.1.2, and C.1.3 configurations:

C.1.1: holes 2 and 4 illuminated (either rewarded),

C.1.2: holes 2 and 14 illuminated (novel rewarded position underlined),

C.1.3: holes 2, 4, 12, and 14 illuminated (novel rewarded position underlined).

Successful responses in C.1.1 or C.1.2 advanced the animal to the next stage. Failing to respond or choosing incorrectly resulted in a 5 s timeout and a restart at C.1.1. In the C.1.3 configuration, all responses—correct or incorrect—reset the sequence to C.1.1 (Figure 2). In C.1.3, any response (correct or incorrect) resulted in a restart at C.1.1. Each rat was provided with the opportunity to complete 10 sequence progressions (C.1.1 to C.1.2 to C.1.3). The C.1.1-to-C.1.3 transition task was administered for two consecutive days.

The ARMIT task assesses free-choice decision making, requiring rats to select among two possible reward locations in the C.1.2 configuration or four in the C.1.3 configuration. The latency between stimulus presentation and the animal’s response serves as an index of decision-making “processing speed.” Event timestamps, automatically logged by the ARMIT software at both stimulus onset and response (or timeout), were used to calculate each rat’s reaction time during task performance.

##### ARMIT Error Classification

In the C.1.3 task, the rats are presented with four response options, of which one is rewarded (correct) and three are not (incorrect). Trial-by-trial classifications were assigned according to the following decision rules: (1) If a trial was correct, no error type was assigned; (2) If a trial was incorrect and the same incorrect choice was made on the immediately preceding trial, the trial was scored as a perseverative error; (3) If a trial was incorrect and the same incorrect choice had been made earlier in the session (but not on the immediately preceding trial), the trial was scored as a regressive error; (4) If a trial was incorrect and the incorrect choice had not been made previously in the session, it was scored as an initial error. For each rat, all trials in the session (maximum 10) were examined sequentially. The number and proportion of perseverative and regressive errors were then calculated for each subject.

##### Sleep Fragmentation and ARMIT Retesting

To isolate the specific effects of sleep fragmentation, we limited participation to high-performing individuals whose baseline performance was stable and near the ceiling. This approach reduces confounding by baseline variability and enhances the interpretability of within-subject performance declines. Six Sham and five GCRsim-exposed rats that had superior ARMIT performance (reward selected at least >20% “above chance”, i.e., >70% in C.1.2, >45% in C.1.3 stages) were subjected to SF using a Sleep Fragmentation Chamber (model 80391; Lafayette Instruments), set to automatically sweep every 2 min to disrupt sleep via tactile stimulation. This protocol has been reported to produce moderate to severe SF [55,56] without significantly reducing overall sleep or impacting sleep macro- or micro-architecture, nor increasing stress hormones [56,57], but can impair cognitive performance [37,40,57].

The rats were placed in the apparatus 75 min before the start of the light period and remained there until ~1 h into the 12 h dark cycle (Zeitgeber T + 1) (~14.25 h). The rats were then reassessed for ARMIT performance immediately after the end of the SF period. The Sleep fragmentation Index (SFI), defined as the ratio of the ATSET performance metrics before and after SF, is an individualized metric of the impact that SF had on ATSET performance. This was calculated for all rats enrolled in the SF study.

### 2.5. Statistical Methods

Data were analyzed using a two-sided Mann–Whitney U test in Prism 10.2 (GraphPad Software, San Diego, CA, USA). For Mann–Whitney analyses, radiation exposure status (Sham vs. 10 cGy GCRsim) served as the independent variable, and the relevant performance metric as the dependent variable. The proportion of rats performing “below chance” in the C.1.1 stage was calculated. Kernel density estimation (KDE) was conducted with the Free Statistics Software package (v1.2.1) [58], using a Gaussian kernel. The resulting distributions were entered into a Risk Reduction Calculator (https://araw.mede.uic.edu/cgi-bin/nntcalc.pl, accessed on 28 May 2025 ) to determine the absolute risk of poor performance attributable to GCRsim exposure.

## 3. Results

Forty-seven rats (26 Sham and 21 GCRsim-exposed) started the HAB phase of the SRT. There were no significant differences in the performance of the Sham and irradiated rats in the SRT stages. However, there was a non-significant trend towards the GCRsim-exposed rats taking more attempts to pass the SRT1F stage (Figure 3). One rat from each of the Sham and GCRsim-exposed cohorts failed to complete the SRT1F stage.

One Sham rat was lost from the study (due to a tail degloving incident) prior to ARMIT testing starting, thus 44 (24 Sham and 20 GCRsim-exposed) rats that successfully completed the SRT1-FP stage progressed onto the ARMIT task.

During the initial five training sessions, rats were exposed exclusively to the C.1.1 configuration, allowing them to form robust memory traces of this task layout. Across all cohorts, performance on the C.1.1 task was high, with each group achieving over 80% correct responses across the 50 trials per session.

Following this familiarization phase, animals were introduced to the C.1.1 → C.1.3 transition paradigm, in which they were rapidly cycled through the C.1.1, C.1.2, and C.1.3 configurations. When re-exposed to the familiar C.1.1 condition, all groups performed comparably, obtaining food rewards in approximately 90% of the ten daily sequences (Figure 4).

In contrast, rats exposed to GCRsim showed markedly poorer performance during the C.1.2 stage. Whereas Sham controls completed approximately 70% of trials successfully (about 20% above chance), GCRsim-exposed animals performed at chance level, consistent with random selection between the two available options (~50% success rate). Kernel density estimation further revealed that roughly 24% of GCRsim-exposed rats performed below chance levels in this stage.

Rats that met the C.1.2 performance criterion advanced to the C.1.3 phase, which introduced four possible response options. In this stage, both Sham and GCRsim-exposed groups selected the rewarded stimulus about half the time—roughly twice the expected chance rate (25%) given a single rewarded option among four alternatives.

We have previously reported that a single night of SF revealed latent ATSET performance decrements in GCRsim-exposed rats that were not manifested under normal testing conditions. We thus selected seven Sham and five GCRsim-exposed rats that had superior performance in the C.1.3 stage, subjected them to SF and reassessed their performance in the ARMIT task the following morning (Figure 5).

The performance of the Sham rats after SF was not significantly different (in any stage) from that seen under normally rested conditions, either at the cohort level (Figure 5) or at the individual level, with the SFI being close to unity in all stages (Figure 6).

While the GCRsim-exposed rats had comparable performance in the C.1.1 and C.1.2 stages under rested and SF conditions, performance in the C.1.3 stage at both the cohort (Figure 5) and individual (Figure 6) levels was significantly reduced. Although the high-performing GCR-exposed rats correctly solved more than 60% of the C.1.3 trials when fully rested, their ability to select the rewarded stimulus (identical to the one used the previous day) after SF dropped to below chance.

There were significant differences in the hole selection strategy of GCRsim-exposed rats in the C.1.3 stage, under rested and SF conditions (Figure 7). While the GCRsim-exposed rats selected the rewarded hole (#12) in more than 50% of the trials when normally rested, after SF exposure, this hole was selected at a significantly (*p* = 0.0286, Mann–Whitney) lower rate. Instead, the SF + GCRsim-exposed rats selected the rewarded hole from the preceding C.1.2 stage (#14) in >60% of trials. In contrast, there were no SF-related changes in the hole selection strategy by the Sham rats.

There were no SF-related changes in the overall incidence or nature of the incorrect choices in Sham rats. However, SF resulted in an increased frequency of incorrect selections in the C.1.3 stage, which was related to a significant increase in perseverative errors (Figure 8), with the rewarded cue from the C.1.2. stage being repeatedly selected.

There were no significant changes in the C.1.3. processing speed after SF in either the Sham s or GCRsim exposed rats (Sham: 3.72 ± 0.60 s. (fully rested) vs. 4.47 ± 1.16 s. (SF); GCRsim: 4.16 ± 0.66 s. (fully rested) vs. SF 5.19 ± 0.52 s. (SF)).

Overall, these data strongly suggest that Sham rats largely maintain behavioral flexibility after SF; however, in contrast, GCR-exposed rats show persistent attentional and decision bias toward outdated cues (impaired cognitive updating). This is a hallmark of behavioral inflexibility, characterized by continued reliance on previously reinforced responses despite new task contingencies.

## 4. Discussion

Decision-making in emergency contexts is a core operational function, requiring the rapid integration of incomplete information, evaluation of competing risks, and execution of time-sensitive actions. Systematic biases or heuristic shortcuts can distort risk assessment and resource allocation, undermining both safety and mission continuity. Thus, the efficiency, accuracy, and impartiality of decision processes are central determinants of resilience in high-stakes environments.

This study identified both overt and latent cognitive deficits in rats exposed to GCRsim using the ARMIT task. Under rested conditions, irradiated animals showed significant impairments in the C.1.2 stage, performing at chance levels while Sham controls achieved ~70% correct responses. Approximately 24% of exposed rats displayed this overt deficit. Because C.1.2 requires updating choices after a shift in reward contingencies, the impairment suggests reduced cognitive flexibility and attentional control, processes dependent on intact prefrontal and corticostriatal circuits.

A subset of irradiated rats exhibited high performance in both C.1.2 and C.1.3 when rested. In these animals, overt deficits were absent at baseline, but vulnerability emerged after one night of SF. Following SF, performance in C.1.3 collapsed, falling below chance despite proficiency the day before. C.1.3 demands higher-level cognitive control to select among four options and resist interference from outdated cues. The performance drop was driven by perseverative errors, reselecting the previously rewarded C.1.2 location. Importantly, this perseverative behavior was specific to irradiated rats; SF had no effect on Sham controls. While the present study demonstrates significant group differences despite relatively small cohort sizes, replication in larger samples would help further confirm the robustness and generalizability of these findings.

From an operational perspective, these findings highlight two distinct modes of cognitive risk for astronauts exposed to chronic low-dose radiation: (1) overt deficits that impair performance even under optimal conditions, and (2) latent deficits that emerge only under stressors such as sleep disruption, leading to abrupt performance collapses in high-demand tasks.

Considerable inter-individual variability in the sensitivity of rats to various SR ions has been observed, 20–36% of rats show overt impairments in attention-dependent tasks after SR exposure alone (“radiosensitive”) [37,38,39,59] Others appear intact under baseline conditions but reveal deficits when challenged by an additional stressor such as SF [37,40,41] or a second SR exposure [60] (“conditionally sensitive”). In most cases, latent deficits emerge only in the most demanding tasks, emphasizing the importance of assessing resilience under stress, not just baseline performance. Similarly, an additional exposure to SR of rats that had no discernible loss of ATSET performance after the first SR dose resulted in set-shifting performance losses [60].

These results support a two-hit model, where SR exposure primes attention-related neural circuits, creating latent vulnerabilities that are unmasked by subsequent stressors such as sleep disruption. In the present study, the frontoparietal/frontostriatal network, which is critical for decision making and set-shifting, appears to be a key locus of this interaction.

The reinforcement learning (RL) framework [61] provides a computational account of decision-making by modeling the trade-off between model-free (habitual) and model-based (goal-directed) behavior, and implicates the corticostriatal circuitry, particularly the dorsolateral striatum and prefrontal cortex, in these processes. Both the medial prefrontal cortex (mPFC) [62,63,64,65] and striatum [66] (in isolation) have been shown to be impacted by SR. Moreover, in our previous study, there was a loss of C.1.3. performance in female rats exposed to 10 cGy GCRsim, suggesting a reduced ability to resist interference from outdated cues [36]. This also appears to occur when female rats are performing in the ATSET task [31]. In both sexes, the additional stress of SF leads to a decreased ability to complete reversal tasks in the ATSET task, again suggesting that the combination of SF and SR impacts the frontoparietal/frontostriatal network [41]. The GCR + SF behavioral profile reflects a shift from model-based (goal-directed) to model-free (habitual) control, where choices are driven by outdated reinforcement histories rather than current contingencies. Increased perseveration suggests a failure to update value associations and suppress obsolete stimulus–reward links.

Under normal conditions, the mPFC modulates the post-parietal cortex (PPC) to shift attention away from outdated cues, while the striatum updates value signals based on new feedback. The PPC can influence striatal learning through attentionally modulated inputs, thereby shaping state-value representations [67]. During decision making, the PPC plays a critical role in representing and updating state-space information, which is essential for both model-free and model-based learning [68]. The PPC transiently encodes distractors or low-level sensory context, whereas the PFC suppresses these signals to maintain working memory or task context. When PFC filtering fails, persistent PPC activity, even if elevated, may act as interference, impairing context-specific decision-making [69]. This process appears to have broken down in GCR-exposed rats subjected to SF.

Mechanistically, GCR exposure may blunt learning rates or reduce reward prediction error signaling, limiting the brain’s ability to revise outdated values. SR exposure disrupts mPFC structure [42], neurochemistry [63], and network integration [65], impairing executive function and biasing decision making. SF likely compounds this by inducing hyper-coherence and attentional rigidity, allowing outdated PPC-driven contextual signals to bias striatal updating. Thus, the combined effects of GCR and SF appear to erode the gating function of the mPFC, resulting in perseverative behavior.

## 5. Conclusions

In summary, these findings provide converging evidence that low-dose SR and SF interact to degrade cognitive flexibility, a process regulated by mPFC–PPC–striatal circuits. SR exposure alone produces overt deficits in attention and set-shifting, while SF unmasks latent vulnerabilities, leading to abrupt behavioral collapse and biased reliance on outdated cues. Our data supports a two-hit model, where SR exposure primes attention-related neural circuits, creating latent vulnerabilities that are unmasked by subsequent stressors (in this study, sleep disruption). The identification of both overt and latent deficits underscores the importance of evaluating not only baseline cognitive capacity but also resilience under operational stressors, with direct implications for Astronaut performance during long-duration deep-space missions.

## Figures and Tables

**Figure 1 life-15-01717-f001:**
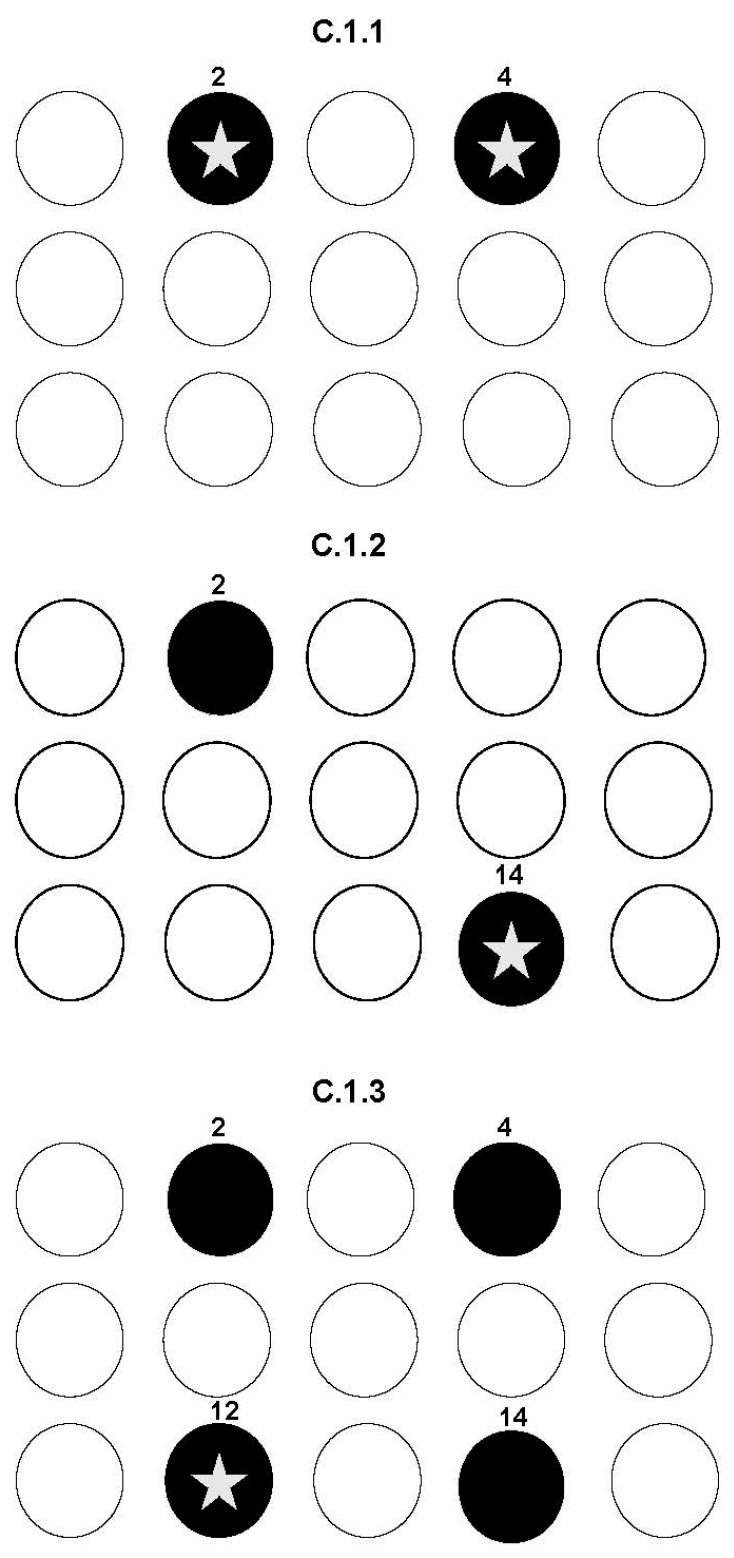
Schematic of ARMIT task configurations. Black circles denote illuminated stimuli, the white star indicates the rewarded hole, and open circles represent unlit positions.

**Figure 2 life-15-01717-f002:**
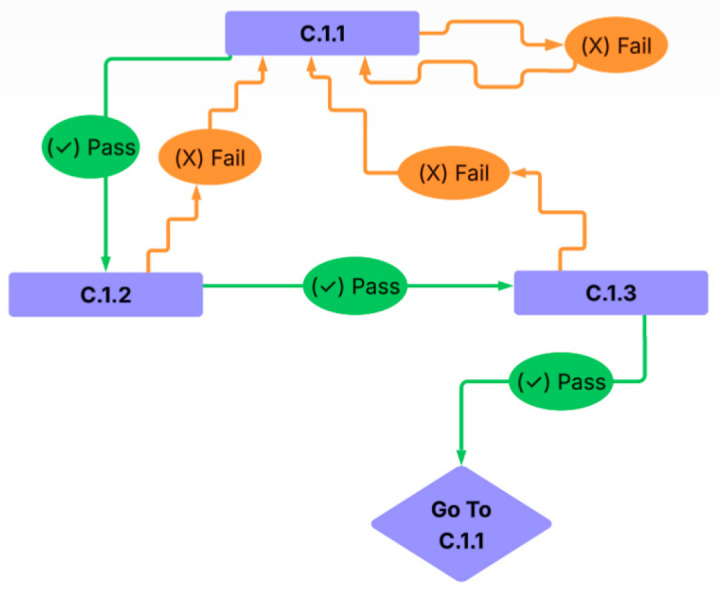
Flow diagram depicting the progressive sequence and Go/No-Go decision points in the ARMIT task.

**Figure 3 life-15-01717-f003:**
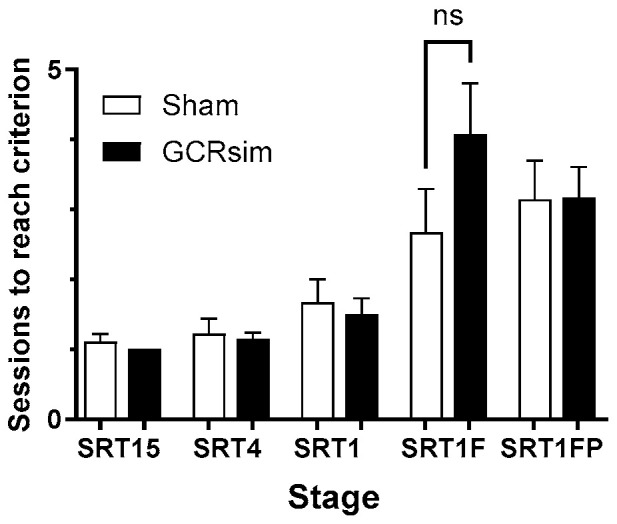
Performance of rats across the stimulus–response training stages. Sham group (open bars) and 10 cGy GCRsim-exposed group (solid bars). Columns show cohort means ± SEM. ns denotes no signficant differences between data sets.

**Figure 4 life-15-01717-f004:**
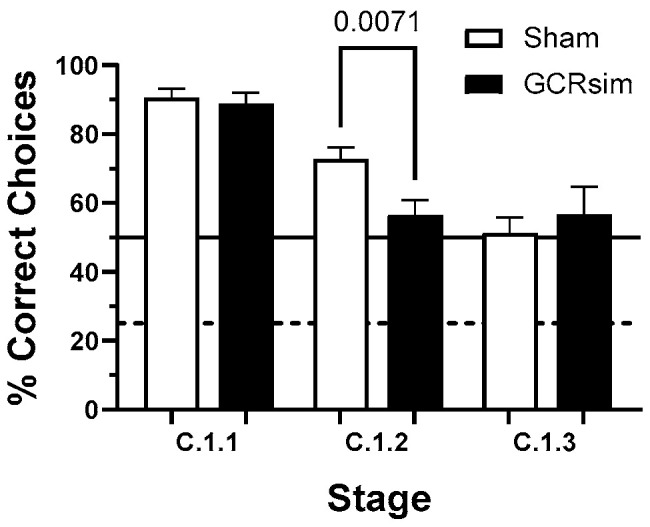
Performance of rats across the C.1.1–C.1.3 stages of the ARMIT task, shown as the percentage of correctly completed trials. Bars indicate group means ± SEM for Sham (open) and 10 cGy GCRsim-exposed (filled) cohorts. Pairwise comparison shows the statistical difference (Mann–Whitney) from Sham. The solid horizontal line represents chance-level performance for the C.1.2 stage, while the dashed line indicates chance-level performance for the C.1.3 stage.

**Figure 5 life-15-01717-f005:**
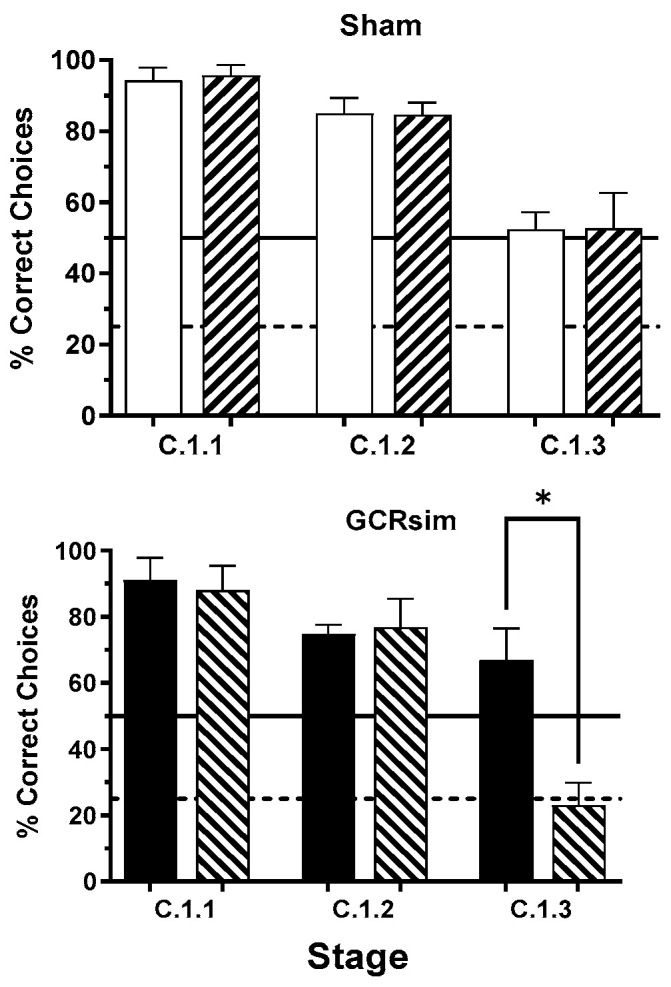
Performance of high performers in the C.1.1 to C.1.3 stages of the ARMIT test under well-rested and sleep-fragmented conditions. Bars represent cohort means ± SEM; Sham (open), 10 cGy GCRsim-exposed (solid), and corresponding sleep-fragmented performance (diagonal stripes). * indicates significant difference from ARMIT performance when fully rested (*p* < 0.05, Mann–Whitney). Horizontal lines denote chance performance in the C.1.2 (solid) and C.1.3 (dashed) stages.

**Figure 6 life-15-01717-f006:**
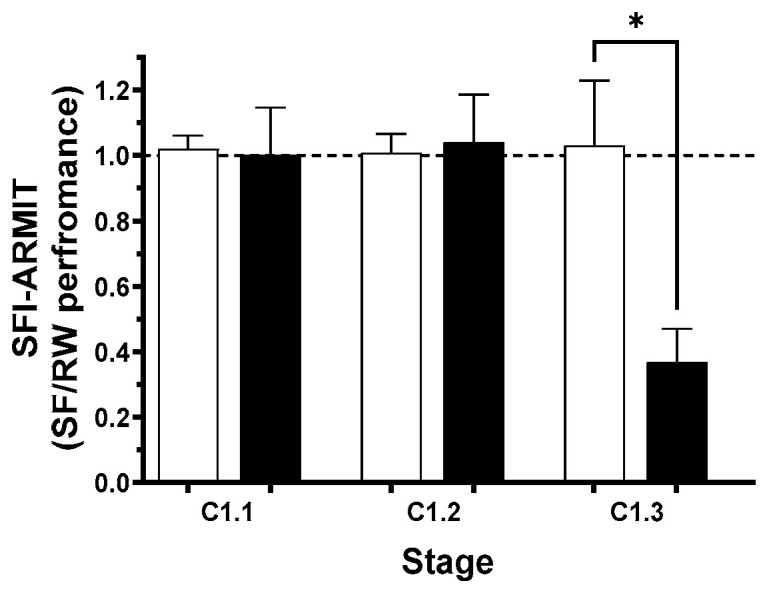
Sleep Fragmentation Index (ratio of post-SF to rested wakefulness performance) for ARMIT performance. Bars represent means ± SEM; Sham (open) and 10 cGy GCRsim (solid). * indicates significant difference from the metrics observed in the Sham rats (*p* < 0.05, Mann–Whitney). The dashed horizontal line denotes SF-neutral performance.

**Figure 7 life-15-01717-f007:**
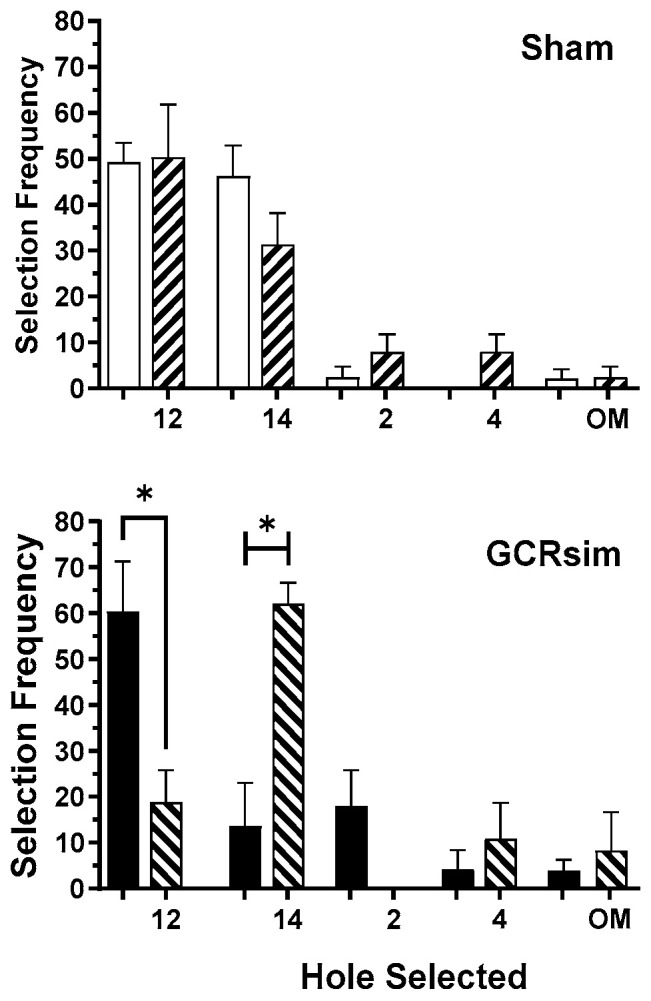
Hole selections (and omissions) by rats during the C.1.3 configuration. Bars represent cohort means ± SEM; Sham (open), 10 cGy GCRsim-exposed (solid), and corresponding sleep-fragmented performance (diagonal stripes). * indicates significant difference from the respective “Rested” selection frequency (*p* < 0.05, Mann–Whitney).

**Figure 8 life-15-01717-f008:**
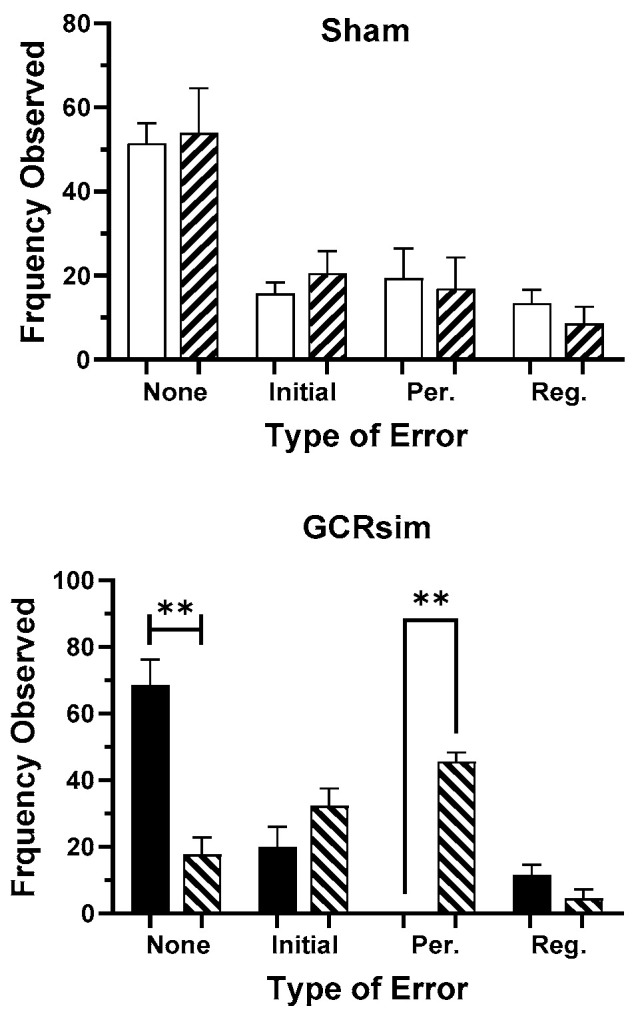
Occurrence and type of errors during the C.1.3 configuration: “Per” indicates perseverative errors, and “Reg” indicates regressive errors. Bars represent cohort means ± SEM; Sham (open), 10 cGy GCR-exposed (solid), and corresponding sleep-fragmented performance (diagonal stripes). ** denotes significant difference from the performance metric obtained from fully rested rats (*p* < 0.01, Mann–Whitney).

**Table 1 life-15-01717-t001:** Experimental timeline (relative to SR exposure).

Event	Time Relative to SR Exposure	Rat Age
Arrival at EVMS	~12 weeks prior	~3 months
Exercise	~11–3 weeks prior	
Shipment to BNL	1 week prior	
SR exposure	-	~6 months
Return to EVMS	1 week post	
Exercise	1–14 weeks post	
ARMIT testing	14–18 weeks post	~10 months

**Table 2 life-15-01717-t002:** Criterion during stimulus response (SRT) training.

Stage ^1^	Response Window	# Correct Selections	Completion Rate	Permitted # Sessions	Days at Criterion
SRT15	N/A	≥30	≥60%	8	1
SRT4	N/A	≥30	≥60%	8	1
SRT1-Timed	30 s	≥30	≥75%	8	2
SRT1-Fast	10 s	≥30	≥75%	10	2

^1^ All sessions in all stages had a maximum duration of 30 min. N/A denotes “Not Applicable”. # denotes “number”.

## Data Availability

The original contributions presented in this study are included in the article. Further inquiries can be directed to the corresponding author.

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
