# Peer review of "A Two-Hit Model of Executive Dysfunction: Simulated Galactic Cosmic Radiation Primes Latent Deficits Revealed by Sleep Fragmentation"

_life, 2025, doi:10.3390/life15111717_

Round 1
Reviewer 1 Report
Comments and Suggestions for Authors
- Line 49: the statement in paratheses (incapacitated crew member rescues) should be inside the sentence
- Line 67: please include citation for ~30% of rats will experience significant loss of performance
Materials and Methods
- Please provide additional details on exercise regimen. Is this forced exercise? During light/dark cycle?
- Why the 3-month delay with testing after SR exposure?
- Rational for including the different criterion for continuation in the study
- Line 203: hole 12? The figure and text do not match
- The different tasks are very confusing/ hard to follow, is the reward hole randomized per trial? Or is the same configuration used for each trial?
- Provide additional rational why only the top performing animals were selected
Results
- Please include a timeline figure to provide clarity in the training, testing and SF experiment
- Line 333: define RW
- Line 291: define SFI
- Figure 4: please clarify statistical comparison, text indicate comparison to sham (which sham group?) but the bar indicate comparison to GCRsim exposed groups
- Figure 5: the line indicates comparison between sham and GCRsim but the text indicate * is to corresponding sham group? Please provide additional text to clarify comparison
- Very low n values -> this should be addressed as a limitation of the study
- Data was shown as average (over the two days) after SF, did the authors see a recovery of any sort on the second day? From the method section, the animals were only in apparatus for 1 day and testing was done over two days
- If animals were only in apparatus for 1 sleep cycle and testing was over 2 days, please show the testing results for each day
Discussion
- Line 378: How does this relate to the current findings? It seems very out of left field especially since this is not defined, citation for SF induces an inflammatory response?
- Line 399: are there other studies to support that SF impact the mPFC-PPC striatal network? The authors have not cited papers on the impact of GCRsim exposure on these network or SF. Would be helpful to provide some more context
- It would be beneficial to see data for animals that were not the best performing one and the impact that SF has on these animals
- A limitation of this study is sex differences, the authors mentioned they’ve previously published data on female but show no female data here
Reviewer 2 Report
Comments and Suggestions for Authors
Review of: A Two-Hit Model of Executive Dysfunction: Simulated Galactic Cosmic Radiation Primes Latent Deficits Revealed by Sleep Fragmentation
By Britten et al.
This paper is clear, concise, and very well written. These studies were conducted by experts in the field and address timely concerns related to potential untoward effects during deep space travel. The touchscreen-based methods are innovative and translational. The cognitive constructs chosen for study are relevant and the analyses among the single and dual stressors are appropriate.
I have very little to offer in the way of suggested improvements. It is clear the authors have spent considerable time optimizing their approach. However, I offer below a few discussion points to consider prior to publication.
Given the inherent complexity of these studies, I do not view the fact that they were only conducted in male subjects as a major problem. And I recognize their previous studies with females examining 10 cGy GCR on task performance. That said, it might be interesting to briefly discuss quantitative and/or qualitative similarities and differences where possible between sexes in the two 10cGy GCR study outcomes.
It appears as if behavioral testing occurred some months after stress exposure. The difficulties of concurrent spaceflight stress/testing are obvious and, to me, do not invalidate the authors’ translational interpretations of in-flight risks and mission success. However, one might also view these outcomes as relevant to astronaut health following return to Earth. It would seem profitable for the authors to speculate on this in the Discussion. After all, although NASA may prioritize risk assessments related to mission success, surely they and most everyone else value longitudinal health outcomes of astronauts after returning home.
Reviewer 3 Report
Comments and Suggestions for Authors
The manuscript by Britten and colleagues describes the effects of GCRsim on performance of the ARMIT task. While irradiated rats showed deficits in C.1.2, a subset of rats experiencing sleep fragmentation displayed additional deficits in C.1.3. When analyzed further, this deficit appeared to consist of perseverative errors, where rats were continuing to choose hole 14 instead of moving to hole 12. While this study is interesting and appears to be well-run, there are some areas for revision noted below.
1) in the results section, the authors state that they chose rats with superior performance for the sleep fragmentation study, but there are not details about how "superior performance" is defined. The authors need to state what criteria were used to select these subjects.
2) The authors argue the the errors shown are a "hallmark of regressive behavior" (page 10, line 337), but then why are these errors labeled as perseverative? The actual labeled regressive errors do not significantly increase. Further, this behavior pattern fits the definition of perseveration, but not regression (moving to an earlier stage of behavior or development as a stress coping mechanism). This needs to be better described and defined consistently throughout.
3) The authors state that no significant differences in processing speed were found, but they did not define how processing speed was measured or how it was analyzed to determine lack of difference.
4) While these data are interesting, it is difficult to draw conclusions from such a small groups of animals (only 5 irradiated "superior performers"). A larger cohort is really need to support a generalized effect of SF an ARMIT performance. It's unclear to me why this separation was needed - the irradiated group was performing above the 25% level so why weren't all of the animals assessed in SF?
Minor comments:
Page 5, line 203 - the text states that hole 11 was used, but the figure states hole 12.
Round 2
Reviewer 1 Report
Comments and Suggestions for Authors
- Very low n values -> this should be addressed as a limitation of the study. This is a relatively small study, but the fact that we have been able to demonstrate significant differences (using a non-parametric test) with such low numbers, actually voids this as a limitation.
- I disagree with this response. There is a lot of variability in animal behavior data and while the authors did find significant difference with this small cohort, a larger cohort would make the results more robust and more likely reproducible.
- Figure 6: There is a floating "A" in the figure on the right
- Define mPFC and PPC before using the acronym in the discussion section
Author Response
- small study, but the fact that we have been able to demonstrate significant differences (using a non-parametric test) with such low numbers, actually voids this as a limitation.
- I disagree with this response. There is a lot of variability in animal behavior data and while the authors did find significant difference with this small cohort, a larger cohort would make the results more robust and more likely reproducible.
We appreciate the reviewer’s comment and agree that larger cohorts can increase confidence in the robustness and reproducibility of behavioral findings. We have added a qualifying statement to acknowledge this consideration in the Discussion. However, given that significant effects were detected despite the small sample size using appropriate non-parametric analyses, we believe the current findings remain valid and informative.
- Figure 6: There is a floating "A" in the figure on the right
Thank you for spotting this. The “A” has been removed.
- Define mPFC and PPC before using the acronym in the discussion section
We appreciate you noticing our oversight. We have now defined these abbreviations and too the opportunity to do some minor rewording of the text associated with those abbreviations.
